# Inflammatory Bowel Diseases: Host-Microbial-Environmental Interactions in Dysbiosis

**DOI:** 10.3390/diseases8020013

**Published:** 2020-05-10

**Authors:** Catherine Colquhoun, Michelle Duncan, George Grant

**Affiliations:** School of Medicine, Medical Sciences and Nutrition, University of Aberdeen, Foresterhill, Aberdeen AB25 2ZD, UK; cate.m.macdonald@googlemail.com (C.C.); michelle.duncan81@gmx.com (M.D.)

**Keywords:** inflammatory bowel diseases, Crohn’s Disease, Ulcerative Colitis, intestinal microbiota, microbial dysbiosis, microbial ecology, microbial colonization

## Abstract

Crohn’s Disease (CD) and Ulcerative Colitis (UC) are world-wide health problems in which intestinal dysbiosis or adverse functional changes in the microbiome are causative or exacerbating factors. The reduced abundance and diversity of the microbiome may be a result of a lack of exposure to vital commensal microbes or overexposure to competitive pathobionts during early life. Alternatively, many commensal bacteria may not find a suitable intestinal niche or fail to proliferate or function in a protective/competitive manner if they do colonize. Bacteria express a range of factors, such as fimbriae, flagella, and secretory compounds that enable them to attach to the gut, modulate metabolism, and outcompete other species. However, the host also releases factors, such as secretory IgA, antimicrobial factors, hormones, and mucins, which can prevent or regulate bacterial interactions with the gut or disable the bacterium. The delicate balance between these competing host and bacteria factors dictates whether a bacterium can colonize, proliferate or function in the intestine. Impaired functioning of NOD2 in Paneth cells and disrupted colonic mucus production are exacerbating features of CD and UC, respectively, that contribute to dysbiosis. This review evaluates the roles of these and other the host, bacterial and environmental factors in inflammatory bowel diseases.

## 1. Background

Crohn’s Disease (CD) and Ulcerative Colitis (UC) are global health problems. Previously considered to be mainly disorders of the developed world, the incidence of these inflammatory bowel diseases (IBD) is now quickly increasing in the developing regions, particularly in areas of rapid urbanization and industrialization [1,2,3]. At present, CD and UC occur at similar frequencies in North America (1:1), but UC predominates (2:1) in Europe and CD (2:1) in Australasia. UC is currently the primary form in Asia, but there are indications of an upswing in CD [2,3]. Most cases (over sixty percent) evidence between 16–25 years of age (adult-onset), with secondary incidence peaks at under 16 years of age (pediatric-onset) and around 55–65 years old (elderly-onset). The nature and severity of the disorders are gender- and age-dependent. Women are most prone to CD, while men are at high risk of UC. Early- (6–10 years), very-early- (2–6 years) and infant- (under two years old) onset forms of IBD are often severe, even at first presentation. Further, in this case, boys have a higher incidence of CD [4,5,6].

UC is primarily a disorder of the large intestine. Inflammation and epithelial disruption begin around the rectum or distal descending colon but may then spread throughout the tissue. Pathology is generally confined to the mucosal layer and, in most cases, terminates at the ileocaecal junction [7]. By contrast, CD can occur in any segment of the gastrointestinal tract, but the most commonly afflicted sites are the lower small bowel (ileum) and the colon. While some lesions are comparable to those with UC, most are penetrative and severe, reaching deep into the sub-epithelium and muscle layer. In extremis, the lesions may lead to the formation of abscesses or fistulas with the mesentery, non-involved intestinal loops, or adjacent organs [8]. Based on pathology, most cases of IBD are readily identifiable as CD or UC. However, there is a proportion of cases in which this is impossible. The description for these is Inflammatory Bowel Disease Unclassified (IBDU) [9].

Predisposition to IBD has been linked to more than 200 gene susceptibility loci, including up to 50 of importance in very-early-onset IBD [10,11]. These loci are associated with an array of metabolic systems, particularly immune development and reactivity, epithelial cell function, gut barrier integrity, and interactions with and control of microbes. Several are disorder-unique, including NOD2 (CARD15), ATG16, and IL2RA for CD and IFNγ, CARD9, and IL-26 for UC [10,11]. Further, polymorphisms in CXCL16 and TLR4 may be significant risk factors for some regions and populations while mutations in IL-10 or IL-10 receptors are associated with severe very-early-onset IBD [12,13]. Despite this, many genetically predisposed individuals never develop IBD, so additional host, lifestyle, or environmental factors have roles, alone or in combination, in the induction of IBD [14].

Intestinal dysbiosis is a feature of CD and UC and is considered a likely triggering or exacerbating factor for both disorders [1,15,16]. However, dysbiosis varies significantly in nature and extent between CD and UC, and even between individuals with the same disorder. Its modes of action and indeed its origins, therefore, remain unclear. The ability of microbes to colonize and persist in the gut is influenced by the host-to-microbe, microbe-to-host, and inter-microbe crosstalk [17,18]. The present review looks at the role of these interactions and environmental factors in IBD (Table 1).

## 2. Microorganisms of the Intestine

Bacteria are the primary microbial population of the intestine: more than 4 × 10^13^ cells, over 1000 different species, with Firmicutes and Bacteroidetes being the most abundant and lesser levels of Actinobacteria and Proteobacteria [19]. In IBD, bacterial abundance and diversity is reduced, especially in CD. The main bacteria affected vary between regions, population groups, and between individuals. However, there is a general pattern for fewer Firmicutes, particularly Clostridial cluster XIVa and IV, and Bacteroidetes, which may be accompanied by an increase in Proteobacteria [15,20,21].

Archaea, viruses, fungi, and protists also feature in the microbiome [22,23]. In IBD, the abundance of *Methanosphaera stadtmanae* increases, the virome enriches [21,24] and fungi such as Basidiomycota, Cystofilobasidiaceae, and Candida increase [21,25]. In contrast, the levels of protist Blastocystis reduce [23].

## 3. Establishment of the Microbiome

The womb has long been considered a sterile environment. However, recent studies indicate that this is not the case. Bacteria are detectable in umbilical cord blood, placenta, fetal membranes and the amniotic fluid of healthy individuals [26,27,28]. These are likely to be the founder components of the fetal intestine microbiota as they are also the main bacteria detected in the very early meconium [26].

Studies in mice suggest that this healthy fetal encounter with bacteria may be a late third-trimester event since pups delivered by hysterectomy two to four days before term are germ-free, but those delivered by cesarean section at full term have microbes in their intestine and meconium [26,29]. Further, in humans, the appearance of bacteria in the amniotic cavity during the second trimester is considered an adverse event and is associated with impaired neonatal outcome, including early or pre-term birth [30,31,32].

The primary colonization of the neonate’s gut occurs at birth, with many strains being acquired from the vaginal microbiome while others originate from maternal skin, colostrum, milk, and the environment. Aerobes predominate initially, but obligate anaerobes supersede them once *Enterobacteriaceae* deplete oxygen from the gut lumen. Within three days, *Bifidobacterium*, *Clostridium,* and *Bacteroides* constitute most of the species present. This microbial balance alters little while milk is the primary food source, but Clostridium, *Akkermansia*, *Bacteroides*, and *Ruminococcuss* capable of digesting complex carbohydrates come to the fore with the switch to solid foods [33,34,35].

During early childhood, the gut microbiome expands rapidly in abundance and diversity, but then progressively stabilizes to an adult-like form between years two and seven. The main features of the adult profile, including a high proportion of Firmicutes and Bacteroidetes, moderate levels of Actinobacteria, and low numbers of Cyanobacteria, Fusobacteria, Proteobacteria, Synergistetes and Verrucomicrobia, are similar worldwide. However, the detailed composition varies regionally and individually according to the traditional diet, lifestyle, and environment [34]. Before stabilization, the microbiome is in flux and highly susceptible to disruption by external factors, such as infection, antibiotics or diet [36,37].

## 4. Influences on Colonization

### 4.1. Diet

The gut microbiome is heterogeneous and highly variable during the first two years of life and greatly influenced by local weaning, dietary, and cooking practices [18,36,38,39,40]. Vegetable-based diets generally favor expansion in Bacteroidetes [2:1] over Firmicutes, whereas animal-product-based diets tend to promote an expansion of Firmicutes [2:1 versus Bacteroidetes] [35]. Once the microbiome stabilizes to an adult-like form, its broad diversity may be little affected by diet, but the abundance and profile of critical bacterial species within it and their associated metabolic activities may alter dramatically. Thus, a switch to high-fat foods rapidly increases the ratio of Firmicutes to Bacteroidetes, while supplemental fiber or prebiotic can at least in part counteract this effect of a high-fat diet [39,41,42,43]. Excessive intake of readily available calories during early life increases the risk of adult IBD [44].

The role of diet in the development of IBD has been extensively studied [45,46,47,48]. The general perception is that a Western-style diet, which contains high levels of animal products and readily digestible carbohydrates but low amounts of vegetable or fruit, predisposes susceptible individuals to IBD [49,50,51]. Several dietary strategies to lower the risk of IBD or to promote recovery or remission have been reported. These include low carbohydrate, low red meat, vegan and gluten-free diets [49,52,53,54,55,56,57,58], but evidence of efficacy is patchy, so a switch to a more varied Mediterranean-type diet remains the general recommended approach.

To date, the primary dietary regime used for the treatment of pediatric CD is exclusive enteral nutrition (EEN) [59,60]. However, partial enteral nutrition in which patients receive a polymeric liquid formula supplement rich in transforming growth factor-β2 (TGF-β2) along with their normal diet has proved effective [61]. EEN acts in part by altering the composition of the gut bacteriome, reducing the abundance of genera that predominate during a flare-up [62]. However, it has little or no ameliorative effect on pediatric or adult UC or adult CD, but may have general nutritional benefits for these patients [63,64].

### 4.2. Antibiotics and Therapeutics

Exposure to antibiotics during pregnancy or early childhood can cause severe disruption to the developing microbiome and an increased risk of dysbiosis and IBD, particularly CD [65,66,67,68]. However, some studies suggest that antibiotics given antepartum or postpartum do not alter IBD risk in offspring [36,69]. Although dysbiosis is a feature of both CD and UC, antibiotic treatment has only limited effects on either disorder. Single antibiotics, such as Metronidazole, or combinations, provide some benefit to CD patients but seem to have little or no benefit to those with UC [70].

Adalimumab and infliximab therapies greatly improve remission rates in IBD. Recent studies indicate that this is in part linked to amelioration of dysbiosis [71,72].

### 4.3. Administered Microbes

Screenings of probiotics show that several have the potential to ameliorate IBD, but their general use in clinical treatment remains limited because of significant inter-individual and inter-population variations in responsiveness to them and indications of side-effects to some strains [73]. Exceptions may be VSL#3 and *E. coli* Nissle 1917, which help promote remission and the prevention of relapse in UC [73].

Fecal microbiota transfer (FMT) has proved a useful treatment for reconstituting the gut microbiome after disruption due to infection or disease. In UC patients, FMT promotes and prolongs remission [74,75]. In addition, there are tentative indications FMT may be of benefit to CD patients [74,75]. Recently, possible live biotherapeutic products, such as *Akkermansia municiphila*, *Bacteroides thetaiotaomicron*, *Roseburia hominis*, and *Faecalibacterium prausnitzii*, have been identified and isolated from intestine contents or the feces of healthy humans [76,77,78,79,80]. These have potential uses in IBD treatment, and several are in early-stage clinical trials. In a recent phase 1 study in teenagers with CD in remission, encapsulated *B. thetaiotaomicron* was well tolerated and had beneficial effects on gut microbial diversity and evenness [81].

### 4.4. Infection

It has long been thought that severe pathogenic infections may increase an individual’s susceptibility to IBD in the long-term, but definitive data is scarce [82]. Indeed, some studies suggest that pathogens, such as *Helicobacter pylori*, may protect individuals from IBD [83]. Nonetheless, some studies have shown that single or repeated bouts of severe gastroenteritis may lead to persistent disruption of the gut microbiome, diminished epithelial integrity, and inappropriate immune responses. These events have been linked to the onset of IBD within one to two years of initial infection [82].

During Salmonella infection, lipopolysaccharide and flagellin bind to the host cell Toll-like receptors 4 and 5, respectively. Interaction of both bacterial components with their receptors initiates down-stream molecular signaling pathways that co-ordinate active host reactions against infection. Disruption of these regulatory pathways adversely affects short- and long-term responses of the host to the pathogen. Thus, C3H/HeN mice exhibit severe gut disruption, weight loss and high mortality when infected with a flagellate (Fla−) *Salmonella enteritidis* (SE). In contrast, they develop a self-limiting infection when dosed with flagellate (Fla+) SE [84]. Mice that survive and recover from the initial Fla− infection develop severe colitis from forty days post-infection while those infected with Fla+ are perfectly healthy in the long-term (Grant, unpublished data). By contrast, C3H/HeJ mice, which have non-responsive TLR4, develop colitis in the long-term after infection with Fla+ but not after exposure to Fla− (Grant, unpublished data).

### 4.5. Cesarean Section

The early gut microbiome of babies delivered by cesarian section differs from that of naturally delivered babies, particularly in the delayed appearance of Bacteroidetes [85]. The dissimilarities persist in the medium term but can be minimized by early seeding of cesarean-born babies with vaginal microbes [85]. There are some associations between cesarean section and increased susceptibility to autoimmune or metabolic diseases, but to date, there is no consistent link to a higher risk of IBD [86,87].

### 4.6. Colostrum and Milk

The early microbiome of breastfed babies generally contain higher levels of Bacteroidetes than Firmicutes, while it is the reverse with formula-fed babies. This difference is due at least in part to their intake of viable bacteria, as well as essential nutrients, immune-growth- and regulatory-factors, present in colostrum and milk [88]. Despite this, there are no clear indications of a higher risk of IBD as a result of formula feeding [14].

Much of the bacteria in breast milk originates from the maternal gastrointestinal tract [89,90]. The route of uptake remains unclear but may involve sampling and the capture of microbes by dendritic cells or macrophages in the intestine and translocation via the mesenteric lymph nodes and lymph and blood circulation to the mammary glands [89,90]. Such a transfer would facilitate seeding of the babies’ intestine with protective or health-promoting microbes, but if the maternal microbiome was imbalanced, as in IBD, it might also facilitate dysbiosis in the baby [91].

Maternal malnutrition adversely affects milk composition, particularly the content of bioactive factors that are critical for the healthy development of the alimentary tract and immune system [18,92]. Babies of severely malnourished mothers often have impaired gut function, and a dysbiosis, which persists even after intensive supplemental nutrition has normalized infant growth [18,92].

## 5. Inter-Bacterial Interactions

Bacteria uses several strategies to identify, inhibit, or eliminate competitors. These include quorum sensing [93] and contact-dependent growth inhibition, in part via type VI secretion systems [94]. Bacteria also secrete bioactive compounds, such as bacteriocins, which are inhibitory or toxic to other bacteria [95]. While most of the secreted factors affect only closely related strains, some act against a broad spectrum of species, including Bacteroidetes and Firmicutes [18,95]. Alternatively, bacteria increase their viability and competitive advantage by utilizing waste metabolites excreted by other microbes. For example, *Roseburia intestinalis* excretes hydrogen, which is a nutrient for *Blautia hydrogenotrophica* that, in return, produces acetate, an important substrate for *R. intestinalis* [17]. This mutualism can extend to great combinations of bacteria that cross-exchange metabolites for their common benefit and is an important factor for the integrity of the healthy microbiome. Loss of keystone species, as in IBD, can disrupt this essential cross-feeding process. The resultant reduction in viability and competitiveness of many bacteria can lead to their loss from the intestine and exacerbation of dysbiosis [96,97].

Biofilms protect bacterial populations within hostile environments. While rare in the healthy intestine, they often associate with inflamed tissue [18,98]. Their composition varies, but with IBD, they generally contain elevated levels of potentially protective bacteria, including *Bacteroides-Prevotella*, *Bacteroides fragilis*, and *Eubacterium rectale-Clostridium coccoides* [98].

## 6. Bacterial–Host Interactions

### 6.1. Fimbriae, Pili, and Lectins

Fimbriae are short, hair-like surface appendages that recognize and bind to host glycoconjugates. They enable bacteria to attach to intestinal mucus or epithelial surfaces in the intestine and thereby facilitate colonization [99,100]. Because of the critical importance of this initial interaction, many bacteria produce fimbriae of differing glycoconjugate-specificities, which may be expressed individually or in combination depending on the host and environmental cues. Indeed, some fimbriae only express when the bacterium is within a living host [101].

Other bacterial surface proteins and soluble lectins augment the actions of fimbriae and aid in adhesion to the gut and the modulation of epithelial cell metabolism [99,100,102].

Adherent-invasive *E. coli* from CD patients have increased expression of long polar fimbriae and point mutations in the FimH adhesin gene [103].

### 6.2. Flagella

Many pathogenic and non-pathogenic bacteria express flagella. In the intestine, they enable the bacterium to move through the lumen and interact with the mucus and epithelial layers [104]. The core protein flagellin is also a primary ligand for Toll-like Receptor 5 and NLRC4 /IPAF. Recognition by host TLR5 initiates a coordinated set of responses. At first, there is acute inflammation in conjunction with the upregulation of cellular survival and repair processes to limit collateral structural damage in the gut. Subsequently, inflammation is downregulated in a controlled manner with the development of adaptive immune responses, if required. A failure in this mixed response can lead to persistent inflammation and severe gut damage [84,105].

Flagellin/TLR5 acts mainly through the Nuclear Factor-κB (NF-κB) pathway. Active NF-κB p65 (RelA) translocates to the nucleus, binds to specific response elements, and initiates transcription of both pro-inflammatory genes, such as IL-8 and CXCL10, which are involved in the initial host responses against the bacterium and counter-regulatory factors, such as TNFAIP3 (A20) and NFKBIA (IκBα), which are essential for subsequent down-regulation of NF-κB and restoration of gut homeostasis [106]. Flagellate commensal and pathogenic bacteria trigger these NF-κB responses. However, inflammation induced by commensals is moderate and rapidly attenuated, which may be due to the early production of counter-regulators, such as A20 [107]. Impairment of this counter-regulation would result in persistent inflammation, which may be a factor in CD since patients often have reduced expression of TNFAIP3 (A20) [108].

Polymorphisms in the TLR5 receptor are a feature in many cases of CD and UC [107,108,109]. The deletion of NLRC4 increases the susceptibility of mice to chemical-induced colitis [105]. Furthermore, flagellin is an antigen detected in CD patients [110]. Moreover, anti-flagellin antibodies limit the expansion of flagellate bacteria in the intestine and ameliorate colitis in an IL-10 knockout model of IBD [111].

### 6.3. LPS and LTA

Lipopolysaccharide is a constituent of the outer membrane of gram-negative bacteria and the primary ligand of TLR4. Lipoteichoic acid is a constituent of the cell wall of gram-positive bacteria and a ligand of TLR2 [100]. The involvement of LPS and LTA in IBD is unclear, but polymorphisms in TLR2 and TLR4 increase host susceptibility to CD or UC [108].

### 6.4. CPS

The bacterial capsule comprises long polysaccharide chains (CPS). These diverse structures provide shielding for surface-expressed antigens or prevent immune-mediated phagocytosis [100]. CPS is also immunomodulatory, albeit the actions are highly strain- or composition-specific. K5 CPS of *E. coli* Nissle 1917 activates TLR5 and potentiates TLR4- and TLR5-reactions to their respective ligands in vitro. In contrast, CPS-expressing *L. rhamnosus* GG attenuated flagellin-TLR5 activation. Furthermore, Polysaccharide A from *Bacteroides fragilis*, delivered in outer membrane vesicles (OMV), promotes the production of Interleukin-10 secreting T-regulatory cells and protects mice against inflammatory colitis [95,100,112].

### 6.5. Peptidoglycan (PG)

These are polymers of alternating β-(1,4)-linked *N*-acetyl-glucosamine and N-acetyl-muramic acid residues with a 3-5 amino acid peptide chain attached to the latter. The peptide chains facilitate cross-linking of the polymers to form a mesh layer out with the plasma membrane of a bacterium. Host cells detect degraded PG fragments via cell surface TLR2. Furthermore, the intracellular receptors NOD1 and NOD2 recognize γ-d-glutamyl-mesodiaminopimelic acid and muramyl dipeptide [100]. Triggering these receptors leads to the release of active NF-κB, the production of pro-inflammatory cytokines, and the up-regulation of cell survival factors [113]. The molecular signaling initiated through NOD2 and TLR may, in part, be inter-dependent. Deficiencies in NOD2, as noted in CD, result in functional dysregulation amongst TLRs and a failure to down-regulate or limit inflammation [113].

### 6.6. Other Bioactives

Some commensal bacteria produce proteins, peptides, or non-proteinaceous compounds that modify molecular signaling pathways in epithelial cells and modulate homeostasis [95,100]. Two soluble proteins (p40 and p75) produced by *L. rhamnosus* GG activate the Akt pathway, suppress apoptosis, and protect epithelial cell layer integrity [100]. Several commensal bacteria are known to block the activity of NF-κB in epithelial cells [114]. For most bacteria, the responsible factors remain unknown, but for *F. prausnitzii* and *B. thetaiotaomicron*, proteins that down-regulate NF-κB have been identified [77,115].

Other bacterial bioactive factors may include soluble lectins that can modulate epithelial cell metabolism [116] and bacteriocins that can influence immune cell cytokine production [117]. Bacterial sphingolipids, particularly of Bacteroides, may have a role in homeostasis. Their levels diminish in IBD, while sphingolipid production by the host increases [118].

### 6.7. SCFA

Short-chain fatty acids (SCFAs) are produced in the colon by bacterial fermentation primarily of poorly digestible polysaccharides. Acetate is the dominant SCFA in feces, followed by propionate and butyrate with formate and succinate present at low concentrations [119,120]. Their production is both bacterial species- and diet-dependent. Acetate is released by many bacteria, while a limited set generates butyrate and propionate. The latter include *Eubacterium rectale*, *E. hallii*, *Faecalibacterium prausnitzii,* and *Ruminococcus bromii*. SCFAs modulate metabolism at various intestinal and systemic sites, but butyrate primarily acts in the lower bowel. It is a preferential energy source for colonic epithelial cells and promotes cell proliferation, tight junction protein interactions, and epithelial layer integrity. Moreover, it induces terminal differentiation and apoptosis in epithelial tumor cells. Butyrate, possibly in conjunction with other SCFAs, can also modulate immune cell function in the lamina propria, blocking the activation of NF-kB and T cell differentiation, enhancing the numbers of IL-10^+^ T regulatory cells and down-regulating responses against commensal bacteria [119,120]. A mixture of SCFAs given topically ameliorated UC and butyrate, in combination with 5-ASA, reduced the severity of 5-ASA refractory UC [120]. Isolated mucosal cells from IBD patients are equally responsive to butyrate as those from the healthy gut but are less receptive when high levels of TNFα are present [121].

### 6.8. Neuromodulators

Intestinal bacteria can produce several neurotransmitters and neuromodulators [122,123]. These include acetylcholine (*Lactobacillus* spp.), dopamine (*Bacillus* spp.), gamma-aminobutyric acid (*Lactobacillus* and *Bifidobacterium* spp.), norepinephrine (*Escherichia* and *Bacillus* spp.) and serotonin (*Streptococcus, Escherichia,* and *Enterococcus* spp.) [122,123,124]. Secreted neurotransmitters from these bacteria may induce host epithelial cells to release factors that regulate neural signaling within the enteric nervous system or may act directly upon it [124,125]. Their actions may occur through the neuro-epithelial link (neuropods) between enteroendocrine cells and neuron and glial cells in the intestine [126]. Intestinal enteroendocrine cells produce an array of hormones, including GLP-1, GLP-2, cholecystokinin, somatostatin, and serotonin, which control many critical aspects of metabolism in the intestine and associated tissues. Gut bacteria can act directly on these cells through TLR- lipopolysaccharide, flagellin, or GpG DNA interactions or indirectly by the release of bioactive metabolites or neuromodulators [124,125,127,128]. Dysbiosis may adversely affect the nature or balance of these neural-enteroendocrine cell communications in IBD.

### 6.9. Segmented Filamentous Bacteria

*Candidatus Savagella* (SFB), a lineage within the Clostridiaceae, interact directly with epithelial cells of the gut by ill-defined mechanisms. They are present in the gut bacteriome of most individuals under three years of age but are below the levels of detection in most adults and the elderly [129]. SFB are potent stimuli of post-natal immune development and maturation. They may also have beneficial roles in colonization resistance and promotion of gut repair but are detrimental for autoimmune arthritis [129]. Furthermore, in small-scale studies, SFB is present in slightly higher numbers on ileocaecal biopsies from UC patients than on control biopsies but are absent from biopsies from CD patients [129,130,131].

## 7. Host–Bacterial Interactions

### 7.1. Antimicrobial Factors

Intestinal cells produce a range of antimicrobial molecules that are important for local gut–microbe interactions and homeostasis [132,133]. Enterocytes can release β-defensins: hBD1 (constitutive), hBD2–hBD6 (inducible), cathelicidin (LL-37/hCAP18), bactericidal permeability-increasing protein (BPI), and chemokine CCL20. In addition to their direct microbiocidal activity, these factors may facilitate innate immune responses through their chemoattractant and immune cell-modulating properties. Resistin-like molecule β (Relmβ) from goblet cells is involved in the protection of the gut barrier [132,133]. Paneth cells produce α-defensins HD5 and HD6, as well as regenerating islet-derived protein 3α (RegIIIα), lysozyme, and phospholipase A2 [133].

The α-defensins HD5 and HD6 are depleted in Crohn’s Disease but up-regulated in Ulcerative Colitis, as are hBD2 and LL-37/hCAP18. However, BPI and CCL20 increase and hBD1 diminishes in CD and UC [132,133,134].

The bactericidal Peptidoglycan Recognition Proteins Pglyrp2, Pglyrp3, and Pglyrp4 are expressed in epithelial cells of GI tract, while Pglyrp1 is mainly present in neutrophils and eosinophils, with low levels in the epithelium [135]. They are essential in the maintenance of homeostasis in the mouse gut bacteriome. In their absence, mice have a reduced abundance of ordinarily dominant bacteria, but a considerable expansion of minor components and increased susceptibility to DSS-induced colitis [136]. Functional Pglyrp3 and NOD2, in combination, are needed to preserve bacterial homeostasis and resistance to DSS-colitis [137]. Polymorphisms in *Pglyrp1*, *Pglyrp2*, *Pglyrp3*, and *Pglyrp4* genes are associated with an increased risk of CD and UC [138].

Reg3α (HIP/PAP) is expressed constitutively in the small intestine, particularly in Paneth cells, but bacteria, infection, and inflammation can trigger increased expression [133,139]. Expression of this bactericidal lectin in the small and large intestine increases with CD and UC. The colonic change is a result of Paneth cell metaplasia [140].

### 7.2. miRNAs

Non-coding RNAs (ncRNAs) are RNA transcripts involved in the control of transcription, stability, or translation of protein-coding genes [141,142]. Of these, microRNAs (miRNAs) are post-transcription regulators of protein-coding genes. They can repress translation of the target mRNA or trigger its degradation and therefore have critical roles in the fine-tuning of epithelial and immune cell development, cell–cell interactions and intestinal and immune function [143,144].

Gut bacteria can differentially modulate the expression of host miRNAs. Indeed, pathogens may use this strategy to limit host immune recognition or reactivity. Non-pathogens such as *Enterococcus faecium* can also modulate intestinal miRNA expression in vivo [145]. There is at present no evidence that bacteria themselves produce miRNAs that can alter host metabolism [145]. However, host miRNAs released into the gut lumen have been shown to alter bacterial gene expression and growth and may influence the composition of the gut bacteriome [146]. Furthermore, maternal milk contains miRNAs [147]. It is, therefore, possible that these influence the initial colonization of the infant intestine and epithelial and immune development.

Environmental and host factors modulate intestinal miRNA expression [148,149,150]. With CD, preferential up-regulation of fifteen miRNAs was evident, particularly of miR130A (linked to suppression of autophagy) [151]. A further eleven miRNAs increased in UC and twelve more elevated in both CD and UC. Furthermore, there is down-regulation of two miRNAs with CD and nine with UC [143,144].

### 7.3. SIgA

The immune system has a significant impact on microbial–mucosal interactions in the gut [152]. Secretory IgA interacts with pathogens, toxins, or harmful antigens in the gut lumen. This adherence of IgA limits their access to the epithelial layer and aids in clearance from the gut [153]. SIgA can also directly modulate the expression of bacterial virulence and bioactive factors, and thereby modulate and shape the bacteriome. Although its primary role is to prevent access to harmful substances, sIgA also has essential functions in aiding capture and transport of antigens across the epithelium to underlying dendritic cells, thereby facilitating the development of specific and appropriate immune responses. Indeed, this sIgA-linked antigen-sampling may be critical in establishing immune tolerance [153,154].

Host sIgA production initiates several months after birth as the immune system develops, and the levels increase steadily during the first few years of life. This change coincides with the rapid expansion and diversification in the microbiota. During early life, the sIgA in milk is thus essential for the protection of the infant from exogenous microbes and antigens. It is likely also to influence the composition of the microbiota that establishes in the gut [153]. Poor maternal nutrition or disease can adversely affect the content and composition of sIgA in milk and hence potentially the composition of the infant microbiome [92].

Recent studies show that only a proportion of the bacteria in the intestine are sIgA-coated: mainly residents that are common to both the small and large intestine, while bacteria exclusive to the colon are generally un-coated [153,155]. Moreover, the amount and affinity of the sIgA coating varied between bacterial species. High-affinity binding may be associated with antigen (such as flagellin)-specific sIgA antibodies [156] and may provide an early indicator to sampling-dendritic cells of the presence of a potentially colitic bacterium [153,155,156].

Higher numbers of highly sIgA-coated bacteria are detectable in feces of IBD patients, in particular unclassified *Clostridiales*, unclassified *Ruminococcaceae*, and *Blautia* species with CD and *Eubacterium dolichum* and *Eggerthella lenta* with UC [156]. Germ-free mice colonized with a mix of these sIgA-reactive strains have a high susceptibility to chemical-initiated colitis [156].

### 7.4. Host Hormones

Some intestinal bacteria express receptors for mammalian hormones, including epinephrine and norepinephrine. Furthermore, gene expression, metabolism, and growth of these bacteria are significantly affected by exposure to the ligands, in vitro and in vivo [123,157].

### 7.5. C-Type Lectin Receptors (CLRs)

Recognition of bacteria, such as *Lactobacillus acidophilus, L. casei*, or *L. reuteri*, by CLRs, initiate the development of T regulatory cells [158]. Surface layer protein A (slpA) is a critical CLR ligand from *L. acidophilus.* When administered orally, it attenuates colitis caused in mice by Dextran Sodium Sulphate [158]. In healthy individuals, the expression of DC-SIGN in the intestinal mucosa is low, but it is high with CD or UC [159].

The expression of Dectin-1 is elevated on immune cells, including macrophages and neutrophils, from CD and UC patients [160]. This change may be due to the modulation of immune responses to bacteria or bacterial factors since secretory IgA-antigen complexes are taken up by intestinal M-cells via the Dectin-1 receptor [154]. After transcytosis, capture by dendritic cells is via DC-SIGN [154]. Antigen sampling and imprinting of dendritic cells in the small intestine also involves a galectin-3/Dectin-1/FcγRIIB receptor complex and may be necessary for establishing tolerance to an antigen.

### 7.6. Glial Cells

CD and UC perturbs the neural complex and glial cell networks [161,162]. In UC, colonic expression of the glial fibrillary acidic protein (GFAP), glial-derived neurotrophic factor (GDNF), and S100 calcium-binding protein β (S100β) increased as did inducible nitric oxide synthase (iNOS), and production of NO. Indeed, the modifications to enteric glial metabolism may have amplified inflammation and led to a loss of neurons [162,163].

Enteric glial cell function alters in CD. The cells are less able to control epithelial layer integrity [164] but have an enhanced capacity to inhibit the proliferation of activated T-cells [165]. Furthermore, the expression of major histocompatibility complex two on ileal glial cells, which is usually very low, is significantly upregulated with CD and many of the cells are associated with T lymphocytes [162,163].

### 7.7. Membrane-Associated Mucins

The glycocalyx of GI epithelial cells comprise a significant part of membrane-anchored cell-surface mucins. There are nine different forms (MUC1, MUC3A, MUC3B, MUC4, MUC12, MUC13, MUC15, MUC16, and MUC17) expressed in combinations on individual cells, and their overall distribution varies throughout the gastrointestinal tract. MUC1 predominates in the stomach, with usually only moderate levels of expression in the small and large intestine. MUC3 is present at high levels in the small intestine and MUC12 in the large intestine. Shedding of the extracellular domain of these mucins occurs in response to inflammatory signals or interaction with a bacterium [166,167].

Colonic expression of these mucins alters with CD and UC [167,168]. In CD, the mRNAs of MUC1, MUC3A, MUC3B, and MUC4 decrease, and MUC13 increases. In UC, MUC1 and MUC13 mRNA increased, and MUC 17 decreased. Moreover, polymorphisms in MUC1 and MUC13 increase the risk of CD and UC, respectively [167].

### 7.8. Bile Acids

During passage down the intestine, many bile acids are converted by bacteria to metabolites that interact with the epithelium, enter the circulation and return to the liver. These bioactive metabolites alter gene expression and function in the intestine systemically [169]. Their nature and actions depend on the compositions of the secreted bile acids and the bacteriome. The latter is, in turn, considerably altered by the often selective anti-bacterial actions of bile acids. Thus, intestinal dysbiosis was evident (*Clostridiales* XIV, *Lachnospiraceae, Rumminococcaceae,* and *Viellonellaceae* reduced, and *Enterobacteriaceae* increased) when the disease altered bile secretions [169,170]. Furthermore, *Firmicute* populations, especially bile acid 7α-dehydroxylating bacteria, are greatly expanded, and *Bacteroidetes* and *Actinobacteria* reduced in rodents given supplemental cholic acid [169] while taurocholic acid increased the abundance of *Bilophila wadsworthia* [171].

Malabsorption of bile acids, associated with a down-regulation of mucosal bile acid transporters, is evident in ileal CD and UC pancolitis, but less so with left-sided UC [172]. The fecal abundance of bacteria expressing 7α-dehydroxylase and 7α-dehydroxysteroid dehydrogenase genes are lowered in UC but increased in CD [173]. The abundance of bacteria expressing bile salt hydrolase genes, such as *Clostridiaceae, Erysipelotrichaceae, Lachnospiraceae,* and *Rumminococcaceae,* was reduced in UC [173].

Disruption of other intestinal transporter and metabolic systems is a factor in CD and UC [174,175]. The impaired expression or functioning of sodium transporters, such as Na^+^/H^+^ exchanger NHE3, dramatically disrupts fluid and salt balance in the intestine [175]. In mice, this loss has led to impaired mucosal integrity, increased inflammation, and dysbiosis, including a diminution in *Clostridiales*, *Lachnospiraceae, and Rumminococcaceae* and expansion of *Bacteriodales, Enterococcaceae*, and *Erysipelotrichaceae* [176]. The deletion of NHE8 leads to impaired mucosal integrity in mice [144].

### 7.9. Oxygen

Strict anaerobic bacteria are adversely affected by oxygen. Disruption of epithelial integrity could allow leakage of systemic oxygen into the gut. An increase in its luminal concentration would repress the growth of strict anaerobes and facilitate an expansion in facultative anaerobes and aerobes in the gut (oxygen hypothesis), as noted in IBD [177]. This was evident in experimental studies in mice in which inflammation and colitis increased colonic epithelial oxygenation and facilitated a major expansion in aerobes, in particular *Escherichia coli*, like that often observed in IBD [178].

## 8. Goblet Cells and UC

### 8.1. Goblet Cells and Mucin

The secretion of mucin, mainly Muc2, by goblet cells, maintains the mucus layers overlying the epithelia of the small and large intestines. In the small intestine, there is a single unattached mucus layer, which moves steadily down the gut due to peristalsis and is replaced continuously with newly secreted mucin. Nonetheless, small transient breaks in the layer can occur because of imbalances between these processes. The large intestine has two mucus layers: a dense and attached inner layer and a loosely associated outer layer. The latter develops through the action of proteases on the inner layer. Bacteria readily attach to, penetrate, and remain within the loose outer mucus layer but are generally excluded from the inner layer [166,179,180]. Physical exclusion is the primary barrier to entry of bacteria into the inner mucus layer [180]. However, the presence of anti-microbial factors that deter the entry of bacteria, particularly flagellate bacteria into the mucin, further enhances its barrier capacity [181]. Other bioactive constituents within mucus layers may include anterior gradient protein two homologs (AGR2), Fc globulin-binding protein (FCG-BP), zymogen granule lectin-like protein 16 (ZG16), resistin-like molecule β (RELMβ), trefoil peptides (TFFs), secretory IgA, and antimicrobial factors released by various enterocytes.

Many factors, including microbes, hormones, and immune factors, modulate goblet cell function, mucus production, and mucin composition [168,180]. Interleukin-10 (IL-10) may indeed be critical for the proper formation of the colonic inner mucin layer since it is permeable to bacteria in IL-10 knockout mice [168,180]. The spread of bacteria into the inner mucin layer is particularly high if the IL-10 knockout mice have elevated TNFAIP3 (A20) [182].

There is a degree of crosstalk between goblet cells and the immune system. Goblet cells sample luminal protein antigens and transfer them preferentially to CD103^+^ dendritic cells, which promote IL-10 and IgA production and development of T regulatory cells [183]. It is presently unclear if this antigen transfer mechanism is involved in Muc2-dependent tolerance-imprinting of dendritic cells. Autophagy-related proteins also modulate the production and release of mucins. Mice with deletions of ATG 5 or ATG 7 have diminished secretion of goblet cell mucins and anti-microbial factors [184].

Most crypts of Lieberkukn are sterile. However, a few crypts in the healthy colon do contain bacteria. This finding has led to the concept of the colonic crypts acting as a reservoir of healthy bacteria for use in the recovery and re-establishment of the colonic bacteriome after trauma [19]. The mode of mucus production in the colon may be critical in the protection of refuge bacteria. The inner mucus layer is formed almost exclusively by mucin released from surface goblet cells. In contrast, goblet cells in the upper crypt contribute to the layer only during periods of trauma, usually by compound secretion of mucin [185].

Significant loss of colonic mucus, discontinuities in the mucus layer, and depletion of goblets cells are features of severe UC and most marked in the descending colon-rectum [180]. The remaining mucus contains an increased proportion of sialomucins and reduced *O*-acetyl sialic acids and sulphomucins, and there is significant bacterial penetration of the inner mucus layer [180]. Moreover, colonic populations of *Desulfovibrio* sp, *Ruminococcus gnavus*, and *R. torques* increase while *Akkermancia muciniphila* decrease. Recent studies found that whipworm therapy ameliorated UC in part because of an NOD-dependent increase in goblet cell mucin production [186,187].

In UC, inflammation occurs exclusively in the colon. However, some studies have highlighted limited histological and physical changes in the ileal mucosa of pediatric UC patients [168,188]. These ileal effects include reduced goblet cell density and mucin production and reduced abundance of *Bacteroides ovatus, Parabacteroides merdae,* and *Alistripes onderdonkii* [168,188].

### 8.2. Regulation of Goblet Cells and UC

Several interdependent mechanisms regulate the production and functional development of intestinal epithelial cells in the small and large intestine. The Wnt/β-catenin pathway modulates proliferation, delineates the dividing cell population within the crypts, and facilitates translocation of Paneth cells to the crypt base. Furthermore, the Hedgehog pathways control cell proliferation, movement, and differentiation within the intestine epithelium, while Bone Morphogenetic Proteins (BMP) stop proliferation and initiate commitment to specific cell lineages [189,190]. The Notch pathway through differential regulation of Hes1 and Atoh1 (Hath1) directs the final commitment of cells to absorptive or secretory lineages. If the Hes1 expression is low and Atoh1 (Hath1) is high, the cells commit to a secretory phenotype. Further transcription factors, including KLF and ELF, act to direct secretory lineage cells to become goblet cells [189,190]. In contrast, when Hes1 is high, while Atoh1 (Hath1) is low, commitment is to an absorptive cell phenotype [189,190].

Colonic expression of Hes1 is high, and Atoh1 (Hath1) is low in UC. Most epithelial cells, therefore, commit to an absorptive cell phenotype, leading to depletion of goblet cells and hence of mucin in the colon [191,192]. Incorrect folding of Muc2 is also evident. This disruption results in endoplasmic reticular stress and persistent activation of NF-κB [189]. Despite mucin depletion, the concentration of olfactomedin 4 in the remaining mucus is substantially increased [193]. Olfactomedin 4 binds β-defensins. These interactions may, therefore, be an attempt to retain β-defensins within the remaining mucus to enhance its barrier capacity [193].

## 9. Goblet Cells and CD

Colonic mucus production and the thickness of the mucin layers do not alter with CD, except around sites of severe lesions. There are some alterations to the glycan structures of the mucin, but these are far less extensive than with UC. The numbers of adherent-invasive *E. coli* associated with the mucin layers increase and many of these express a mucus-degrading protease whose activity enables them to pass through the inner mucus layer and interact directly with epithelial cells [194]. Goblet cell density, Muc2, Trefoil Factor 2 (TFF2), and TFF3 expression is low in the terminal ileum of pediatric CD patients, but not in the colon [168].

## 10. Paneth Cells and CD

Paneth cells are one of the four lineages derived from rapidly dividing mix cells in the lower crypt. As these cells move up the crypt and stop dividing, the cells commit to a specific lineage; mature and absorptive, goblet and endocrine cells continue upwards onto the villus surface. In contrast, the Paneth cells return to the base of the crypt in part under the guidance Wnt /TCF4, which is activated by agonists released from sub-epithelial myofibroblasts. Once in the crypt base, the mature Paneth cluster around four to six intestinal stem cells. They produce and secrete a range of anti-microbial and bioactive factors, including α-defensins HD5 and HD6, RegIIIα (HIP/PAP), lysozyme and phospholipase A2 as well as EGF, Wnt3 and Notch ligand [189]. Some of these bioactives protect and control the progenitor cells while others have more extensive roles in shielding the host against pathogens and in shaping the gut microbiome. NOD2, an intracellular receptor in Paneth cells, regulate production and secretion of these factors’ secretory processes, possibly acting in tandem with the cell-surface receptor TLR2 [189,195,196]. Symbiosis requires the proper functioning of NOD2, and protein kinase LRRK2 [197] and is lost if autophagy in Paneth cells is impaired [198].

Loss of function variants of NOD2 altered activity of the autophagy factor ATG16L1, and reduced action of TCF4 (TCF7L2) of the Wnt pathway in Paneth cells are features of high risk for ileum CD. Moreover, functional disruption of endosomal stress protein XBP1, Toll-like receptor 9, calcium-mediated potassium channel KCNN4, immunity-related GTPase family M protein IRGM, and cytosolic protein kinase LRRK2 can be evident. The overall effect is to attenuate production or release of anti-microbial factors, such as HD5, HD6, and lysozyme by ileal Paneth cells [134,189,197]. In contrast, while absent in the healthy colon, HD5 and HD6 are found in the distal colon of CD patients. This change may be due to the appearance of Paneth-like cells in the distal colon [140].

Paneth cell pathology and dysbiosis dictate the likely severity of CD, and there are two classifications: phenotype one where more than twenty percent of Paneth cells are of abnormal appearance; and phenotype two in which less than twenty percent of Paneth cells are of abnormal appearance [199].

Phenotype one is linked with particularly severe CD, and a high risk of relapse after surgery [199]. It evidences in around twenty percent of adult CD patients and forty-five percent of pediatric CD patients [199,200]. Surprisingly, while impairment of NOD2 and ATG16L1 is an apparent feature of adult phenotype one CD, this linkage is less evident in pediatric phenotype one CD [200]. Phenotype one is associated with a low ileal abundance of *Anaerostipes, Blautia, Faecalibacterium,* and *Ruminococcaceae*, but elevated *Corynebacterium and Erysipelotrichaceae* [199,200].

Phenotype two is indicative of a more moderate form of CD and is associated with a low ileal abundance of *Corynebacterium* and *Erysipelotrichaceae*, but high levels of *Lachnospira, Odoribacteraceae, Peptostreptococcus*, *and Porphyromonas* [199,200].

## 11. Dysbiosis: Cause or Consequence

A well-established characteristic of inflammatory bowel diseases is a reduction by twenty-five or more percent in the diversity and richness of the intestine microbiota. The exact changes cannot be readily defined because they vary between CD and UC, with region and population group, and between individuals. In general, potentially beneficial organisms such as *Lachnospiraceae* (Firmicutes), *Bifidobacterium* species (Actinobacteria), *Roseburia* (Firmicutes), *Sutterella* (Proteobacteria), and *Faecalibacterium prausnitzii* (Firmicutes) are greatly reduced whereas pathogens/pathobionts, including Proteobacteria, Fusobacteria species, and *Ruminococcus gnavus* are elevated [201,202,203,204].

Despite extensive study, the underlying reasons for dysbiosis remain unclear [205]. It may be a result of a lack of exposure to critical protective microbes during early formation of the intestine microbiota or the failure of protective bacteria to find a niche in the colon due to adverse host responses against them from the host. This may lead to impaired development of host intestine and immune systems and long-term intestinal dysfunction. However, since dysbiosis does not lead to IBD in all individuals it is likely that host factors that predispose to the development of IBD play a role in onset. These may facilitate an expansion of pathobionts in the intestine, which could in turn lead to metabolic changes in the gut that limit the viability or competitiveness of potentially beneficial bacteria [206,207,208,209,210].

Irrespective of the its origins, dysbiosis has a major involvement in both CD and UC. It exacerbates the severity, duration and frequency of inflammatory episodes in the intestine. Treatments that address dysbiosis and promote eubiosis can greatly reduce the gravity of these inflammatory disorders, although they do not cure them.

## 12. Conclusions

CD and UC are severe diseases in which chronic intestinal inflammation and severe tissue disruption may be caused or exacerbated by intestinal dysbiosis. Reduced abundance and diversity of the microbiome may arise from lack of exposure to important protective commensal microbes and overexposure to competitive pathobionts during early life (hygiene hypothesis). Alternatively, many commensal bacteria may not find a suitable intestinal niche or may fail to proliferate or function in a protective/competitive manner if they do colonize. This in turn may also allow an expansion in opportunistic pathobionts.

Bacteria express a range of factors, such as fimbriae, flagella, and secretory bioactive compounds that enable them to attach to mucus or epithelium, modulate local gut metabolism, and outcompete other species. However, the host also releases factors, such as secretory IgA, antimicrobial factors, hormones, and mucins, which can prevent or regulate bacterial interactions with the gut or disable the bacterium by blocking the expression or action of fimbriae, flagella, or other factors. Bacteria unaffected by these host factors or able to counteract them persist in the gut, whereas those that cannot are unable to colonize. Alternatively, they may establish in the intestine but fail to proliferate or function in a protective/competitive manner.

Dysbiosis has multiple origins and exacerbating factors. In CD, impaired functioning of NOD2 in Paneth cells is a critical aspect. The resultant reduction in production of anti-microbial and bioactive compounds and loss of molecular signaling weakens regulatory control over bacteria in the gut. In UC, disruption of the Notch or IL-10 regulatory pathways impair mucus production and facilitates access of pathobionts to the colonic epithelium. 

To promote and maintain eubiosis in the gut and ameliorate CD and UC requires a combination of therapies. These treatments may need to be population- or individually targeted and involve administration of pure cultures of potentially beneficial bacteria along with supplementary treatments that facilitate the functional survival and colonization in the gut.

## Figures and Tables

**Table 1 diseases-08-00013-t001:** Bacterial and host factors with possible involvement in bacterial/intestine crosstalk in Crohn’s Disease and Ulcerative Colitis.

Factor	CD	UC	Reference
Bacterial factors			
Fimbriae	√	-	103
Flagella/TLR5	√	-	110
Lipopolysaccharide (LPS)/TLR4	-	√	108
Lipoteichoic acid (LTA)/TLR2	√	√	108
Capsule polysaccharides	?	?	95
Peptidoglycan/TLR2	√	-	113
Peptidoglycan/NOD2	√	-	113
Released bioactives	?	?	95
Short chain fatty acids (SCFA)	-	√	120
Neuromodulators	?	?	123
Biofilms	√	√	97
**Host factors**			
Intestinal transporter systems	√	√	172
Dendritic/immune cells	√	√	201
Secretory IgA (sIgA)	√	√	156
Antimicrobial factors	√	√	134
Peptidoglycan recognition proteins	√	√	138
Toll-like receptors (TLRs)	√	√	108
NOD2	√	-	113
microRNAs (miRNAs)	√	√	143
Hormones	?	?	123
C-type lectin receptors	√	√	154
Glial cells	√	√	162
Epithelial cell surface mucins	√	√	168
Goblet cells/mucin	*	√	180
Mucin layer integrity	*	√	180
Paneth cell phenotype	√	-	198
Paneth cell NOD2	√	-	134
Paneth cell antimicrobials	√	-	134
Paneth cell autophagy	√	-	198

√ known. - none known. ? unknown or unclear. * minor involvement compared to Ulcerative Colitis (UC), possible role in pediatric Crohn’s disease (CD). NOD2, Nucleotide-binding oligomerization domain-containing protein 2.

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
