# Peer review of "Inflammatory Bowel Diseases: Host-Microbial-Environmental Interactions in Dysbiosis"

_diseases, 2020, doi:10.3390/diseases8020013_

Round 1

Reviewer 1 Report

  • “CD originates in the terminal ileum and the proximal ascending colon”

This is false. CD could interest only the colon or only the upper small bowel.

  • “Within three days, Bifidobacterium, Clostridium, and Bacteroidetes constitute most of the species present.”

Write in italics the name of the specific species. Do not confuse the name of the Phlyla with the name of the species

  • “Bacteroides, Clostridium and Ruminococcus and other strains”

Bacteroides, Clostridium and Ruminococcus are not strains but genus

  • “To date, the only dietary regime successfully used for the treatment of pediatric CD is exclusive

enteral nutrition”

Cite the role of “Modulen”

  • “One exception may be VSL#3, which helps promote remission and the prevention of relapse in UC”

Cite the data regarding E. coli Nissle 1917 in maintaining remission in UC

  • “It has long thought that severe pathogenic infections may increase an individual's susceptibility to IBD in the long-term, but definitive data is scarce”

Cite the possible preventive effect of chronic infection, like H. pylori, on develop of chronic inflammatory bowel diseases (On Inverse Association Between Helicobacter pylori Gastritis and Microscopic Colitis: The European Data. Ribaldone DG, Simondi D, Astegiano M, Pellicano R.

Inflamm Bowel Dis. 2016 Mar;22(3):E11-2.)

  • Deepen the topic about typical dysbiosis in IBD is a reduction of diversity (about 30%)

  • Deepen the topic about if dysbiosis is a cause or a consequence of the inflammation

  • Cite the role of FMT in IBD (for example: Newman KM, Vaughn BP. Efficacy of intestinal microbiota transplantation in ulcerative colitis: a review of current literature and knowledge. Minerva

Gastroenterol Dietol. 2019 Dec;65(4):268-279. doi: 10.23736/S1121-421X.19.02610-2. Epub 2019 Jul 24. PubMed PMID: 31347341.)

  • Cite the effect of the therapy on dysbiosis in IBD (for example: Adalimumab Therapy Improves Intestinal Dysbiosis in Crohn's Disease. Ribaldone DG, Caviglia GP, Abdulle A, Pellicano R, Ditto MC, Morino M, Fusaro E, Saracco GM, Bugianesi E, Astegiano M. J Clin Med. 2019 Oct 9;8(10). pii: E1646. doi: 10.3390/jcm8101646.)

Author Response

Thank you for your helpful comments and queries. I hope that I have addressed them satisfactorily.

“CD originates in the terminal ileum and the proximal ascending colon”

This is false. CD could interest only the colon or only the upper small bowel.

Ln 43-46               Text amended to read “CD can occur in any segment of the gastrointestinal tract, but the most commonly afflicted sites are the lower small bowel (ileum) and the colon”

Reference 8 replaced with Roda, G., Chien Ng, S. et al. Crohn’s disease. Nature Reviews Disease Primers 2020. 6 (1), art. no. 22. doi:10.1038/s41572-020-0156-2

“Within three days, Bifidobacterium, Clostridium, and Bacteroidetes constitute most of the species present.”

Write in italics the name of the specific species. Do not confuse the name of the Phlyla with the name of the species

Ln 96      Corrected

“Bacteroides, Clostridium and Ruminococcus and other strains”

Bacteroides, Clostridium and Ruminococcus are not strains but genus

Ln 97-99 Corrected “is the primary food source, but Clostridium, Akkermansia, Bacteroides, and Ruminococcus capable of digesting complex carbohydrates come to the fore with the switch to solid foods

“To date, the only dietary regime successfully used for the treatment of pediatric CD is exclusive enteral nutrition”

Cite the role of “Modulen”

Ln 127-130           New text “To date, the primary dietary regime used for the treatment of pediatric CD is exclusive enteral nutrition [EEN] [59, 60 However, partial enteral nutrition in which patients receive a rich in TGF polymeric liquid formula supplement along with their normal diet has proved effective [61].

New reference 61

“One exception may be VSL#3, which helps promote remission and the prevention of relapse in UC”

Cite the data regarding E. coli Nissle 1917 in maintaining remission in UC

Ln 149-151 Corrected. “Exceptions may be VSL#3 and E. coli Nissle 1917, which help promote remission and the prevention of relapse in UC [73-75]”.

“It has long thought that severe pathogenic infections may increase an individual's susceptibility to IBD in the long-term, but definitive data is scarce”

Ln 164-165 Corrected. “but definitive data is scarce [82]. Indeed, some studies suggest that pathogens, such as Helicobacter pylori, may protect individuals from IBD [83]. Nonetheless, some”

Ref 83 replaced with “Kayali, S., Gaiani, F., Manfredi, M. et al. Inverse association between helicobacter pylori and inflammatory bowel disease: Myth or fact? Acta Biomedica 2018. 89, pp. 81-86”.

Cite the role of FMT in IBD (for example: Newman KM, Vaughn BP. Efficacy of intestinal microbiota transplantation in ulcerative colitis: a review of current literature and knowledge. Minerva Gastroenterol Dietol. 2019 Dec;65(4):268-279. doi: 10.23736/S1121-421X.19.02610-2. Epub 2019 Jul 24. PubMed PMID: 31347341.)

Ln 152-155           Added “Fecal microbiota transfer (FMT) has proved a useful treatment for reconstituting the gut microbiome after disruption due to infection or disease. In UC patients, FMT promotes and prolongs remission [74, 75]. In addition, there are tentative indications FMT may be of benefit to CD patients [74, 75].

Replaced references 74 and 75 with: Newman, K.M., Moscoso, C.G., Vaughn, B.P. Fecal microbiota transfer and inflammatory bowel disease: A therapy or risk? (2019) Microbiome and Metabolome in Diagnosis, Therapy, and other Strategic Applications, pp. 425-434. DOI: 10.1016/B978-0-12-815249-2.00045-2 and Imdad, A., Nicholson, M.R., Tanner-Smith, E.E., Zackular, J.P., Gomez-Duarte, O.G., Beaulieu, D.B., Acra, S. Fecal transplantation for treatment of inflammatory bowel disease (2018) Cochrane Database of Systematic Reviews, 2018 (11), art. no. CD012774. DOI: 10.1002/14651858.CD012774

Cite the effect of the therapy on dysbiosis in IBD (for example: Adalimumab Therapy Improves Intestinal Dysbiosis in Crohn's Disease. Ribaldone DG, Caviglia GP, Abdulle A, Pellicano R, Ditto MC, Morino M, Fusaro E, Saracco GM, Bugianesi E, Astegiano M. J Clin Med. 2019 Oct 9;8(10). pii: E1646. doi: 10.3390/jcm8101646.)

Ln 141-145 amended. “Adalimumab and infliximab therapies greatly improve remission rates in IBD. Recent studies indicate that this is in part linked to amelioration of dysbiosis [71, 72]”.

New references 71, 72. Ribaldone DG, Caviglia GP, Abdulle A. et al. Adalimumab Therapy Improves Intestinal Dysbiosis in Crohn's Disease. J Clin Med. 2019. 8(10). pii: E1646. doi: 10.3390/jcm8101646

Dovrolis, N., Michalopoulos, G., Theodoropoulos, G.E. et al. The interplay between mucosal microbiota composition and host gene-expression is linked with infliximab response in inflammatory bowel diseases. Microorganisms 2020. 8 (3), art. no. 438. DOI: 10.3390/microorganisms8030438

Deepen the topic about typical dysbiosis in IBD is a reduction of diversity (about 30%). Deepen the topic about if dysbiosis is a cause or a consequence of the inflammation

Additional text Ln 585-607

  1. Dysbiosis: cause or consequence

A well-established characteristic of inflammatory bowel diseases is a reduction by twenty-five or more percent in the diversity and richness of the intestine microbiota. The exact changes cannot be readily defined because they vary between CD and UC, with region and population group, and between individuals. In general, potentially beneficial organisms such as Lachnospiraceae (Firmicutes), Bifidobacterium species (Actinobacteria), Roseburia (Firmicutes), Sutterella (Proteobacteria), and Faecalibacterium prausnitzii (Firmicutes) are greatly reduced whereas pathogens/pathobionts, including Proteobacteria, Fusobacteria species, and Ruminococcus gnavus are elevated [202-204].

Despite extensive study, the underlying reasons for dysbiosis remain unclear [205]. It may be a result of lack of exposure to critical protective microbes during early formation of the intestine microbiota or failure of protective bacteria to find a niche in the colonise due to adverse host responses against them from the host. This may lead to impaired development of host intestine and immune systems and long-term intestinal dysfunction. However, since dysbiosis does not lead to IBD in all individuals it is likely that host factors that predispose to the development of IBD play a role in onset. These may facilitate an expansion of pathobionts in the intestine, which could in turn lead to metabolic changes in the gut that limit the viability or competitiveness of potentially beneficial bacteria [206-210].

Irrespective of the its origins, dysbiosis has a major involvement in both CD and UC. It exacerbates the severity, duration and frequency of inflammatory episodes in the intestine. Treatments that address dysbiosis and promote eubiosis can greatly reduce the gravity of these inflammatory disorders, although they do not cure them.

202        Alam MT, Amos GCA, Murphy ARJ.et al. Microbial imbalance in inflammatory bowel disease patients at different taxonomic levels. Gut Pathog. 2020 12:1. doi: 10.1186/s13099-019-0341-6.

203         Zareef, R., Younis, N., Mahfouz, R. Inflammatory bowel disease: A key role for microbiota? Meta Gene 2020. 25, art. no. 100713. DOI: 10.1016/j.mgene.2020.100713

204         Basso PJ, Câmara NOS, Sales-Campos H. Microbial-Based Therapies in the Treatment of Inflammatory Bowel Disease - An Overview of Human Studies. Front Pharmacol. 2019. 9:1571. doi: 10.3389/fphar.2018.01571

205         Khan, I., Ullah, N., Zha, L. et al. Alteration of gut microbiota in inflammatory bowel disease (IBD): Cause or consequence? IBD treatment targeting the gut microbiome. Pathogens 2019. 8 (3), art. no. 126. DOI: 10.3390/pathogens8030126

206         Agrawal, G., Clancy, A., Huynh, R., Borody T. Profound remission in Crohn’s disease requiring no further treatment for 3–23 years: a case series. Gut Pathog. 2020. 12:16 doi.org/10.1186/s13099-020-00355-8.

207         Diener, C., Gibbons, S.M., Resendis-Antonio, O. MICOM: Metagenome-scale modeling to infer metabolic interactions in the gut microbiota mSystems 2020. 5 (1), art. no. e00606-19. DOI: 10.1128/mSystems.00606-19 1.     

208         Kiely CJ, Pavli P, O'Brien CL. The role of inflammation in temporal shifts in the inflammatory bowel disease mucosal microbiome. Gut Microbes 2018 9(6):477-485. doi: 10.1080/19490976.2018.1448742.

209         Larabi A, Barnich N, Nguyen HTT. New insights into the interplay between autophagy, gut microbiota and inflammatory responses in IBD. Autophagy 2020 16(1):38-51. doi: 10.1080/15548627.2019.1635384.

210         Zheng, C., Huang, Y., Ye, Z., et al. Infantile onset intractable inflammatory bowel disease due to novel heterozygous mutations in TNFAIP3 (A20). Inflammatory Bowel Diseases 2018 24 (12), pp. 2613-2620. DOI: 10.1093/IBD/IZY165

Reviewer 2 Report

The review article successfully captures the important interactions between host, microbes and the environment in Inflammatory Bowel Diseases. I have following questions/concerns.

1) Under section 5, Inter-bacterial interactions, could authors add additional information about role of cross feeding in UC/CD.

2) Under section 7, Host-bacterial interactions, could authors expand the effect of oxygen on colonization of species leading to worst outcomes of UC/CD.

3) Please check typos at lines 16, 31, 77 and 209

Author Response

Thank you for your helpful comments. Hopefully, I have now included sufficient additional information.

1)Under section 5, Inter-bacterial interactions, could authors add additional information about role of cross feeding in UC/CD.

Ln 214-217           Amended. “This mutualism can extend to great combinations of bacteria that cross-exchange metabolites for their common benefit and is an important factor for the integrity of the healthy microbiome. Loss of keystone species, as in IBD, can disrupt this essential cross-feeding process. The resultant reduction in viability and competitiveness of many bacteria could lead to their loss from the intestine and exacerbation of dysbiosis [97, 98]”.

New references 97, 98.  Evans, C.R., Kempes, C.P., Price-Whelan, A., Dietrich, L.E.P. Metabolic Heterogeneity and Cross-Feeding in Bacterial Multicellular Systems. Trends in Microbiology. DOI: 10.1016/j.tim.2020.03.008 and Gutiérrez, N., Garridoa, D. Species deletions from microbiome consortia reveal key metabolic interactions between gut microbes. mSystems, 2019. 4 (4), art. no. e00185-19. DOI: 10.1128/mSystems.00185-19

2) Under section 7, Host-bacterial interactions, could authors expand the effect of oxygen on colonization of species leading to worst outcomes of UC/CD.

Ln 474-476           Additional text: “This was evident in experimental studies in mice in which inflammation and colitis increased colonic epithelial oxygenation and facilitated a major expansion in aerobes, in particular Escherichia coli, like that often observed in IBD [178]”.

New reference 178          Cevallos, S.A., Lee, J.-Y., Tiffany, C.R. et al. Increased epithelial oxygenation links colitis to an expansion of tumorigenic bacteria mBio, 2019. 10 (5), art. no. e02244-19. DOI: 10.1128/mBio.02244-19

3) Please check typos at lines 16, 31, 77 and 209 (now 223)

Corrected